# India based neutrino observatory, physics reach and status report

**D. Indumathi⋆**

The Institute of Mathematical Sciences,
Chennai and Homi Bhabha National Institute, Mumbai

⋆ indu@imsc.res.in

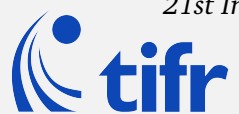

*21st International Symposium on Very High Energy Cosmic Ray Interactions
(ISVHECRI 2022)
Online, 23-28 May 2022*

## Abstract

The India-based Neutrino Observatory (INO) is a proposed underground facility located in India that will primarily house the magnetised Iron CALorimeter (ICAL) detector to study atmospheric neutrinos produced by interactions of cosmic rays with Earth's atmosphere. The physics goal is to to make precision measurements of the neutrino mixing and oscillation parameters through such a study. We present here the results from detailed simulations studies, as well as a status report on the project. In particular, we highlight the sensitivity of ICAL to the open issue of the neutrino mass ordering, which can be determined *independent of the CP phase* at ICAL.

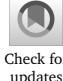

# 1 Introduction

The possibility of neutrino oscillations was originally suggested by Pontecorvo (actually, he examined the possibility of neutrino anti-neutrino oscillations) [1]. The theory of neutrino mixing in three-flavours was given by Maki, Nakagawa and Sakata [2] and the effect of matter on the propagation of neutrinos was discussed by Wolfenstein [3], Mikheev and Smirnov [4]. It is now well-established that neutrino flavours mix, and hence neutrino oscillations have been observed in diverse instances such as in solar neutrinos, atmospheric neutrinos, reactor neutrinos, etc.; see Ref. [5] for a historical review. Neutrino mixing in three flavours is parametrised by the PMNS matrix, $U_{PMNS}$, given by

$$U_{PMNS} = \begin{pmatrix} c_{12}c_{13} & s_{12}c_{13} & s_{13}e^{-i\delta_{CP}} \\ -c_{23}s_{12}-s_{23}s_{13}c_{12}e^{i\delta_{CP}} & c_{23}c_{12}-s_{23}s_{13}s_{12}e^{i\delta_{CP}} & s_{23}c_{13} \\ s_{23}s_{12}-c_{23}s_{13}c_{12}e^{i\delta_{CP}} & -s_{23}c_{12}-c_{23}s_{13}s_{12}e^{i\delta_{CP}} & c_{23}c_{13} \end{pmatrix}. \tag{1}$$

Here $c_{12} = \cos\theta_{12}$, $s_{12} = \sin\theta_{12}$ etc., and $\delta_{CP}$ denotes the CP violating (Dirac) phase. By definition, the $3 \times 3$ neutrino mass matrix $M_\nu$ is diagonalised in the charged-lepton mass basis. The parameters involved are the mixing angles $\theta_{ij}$ and the mass-squared differences $\delta_{ij} \equiv m_i^2 - m_j^2$. A combined analysis [6] of the data from various experiments indicates that the across-generation mixing angle $\theta_{13} \sim 8.5°$ is smaller than $\theta_{12} \sim 34°$, $\theta_{23} \sim 45°$, while the CP phase is still not well-established. There are two independent mass squared differences, $|\delta_{31}| \gg \delta_{21}$. While $\delta_{21} \sim 7.6 \times 10^{-5}$ eV$^2$ is known to be positive, the *sign* of the other mass squared difference is unknown, with $|\delta_{31}| \sim 2.5 \times 10^{-3}$ eV$^2$, with the sum of all the neutrino masses constrained by cosmological considerations to be less than about 2 eV. One of the important open issues to be settled by future experiments is the so-called neutrino mass ordering, *viz.*, whether the third mass eigenstate is heavier or lighter than the others. We will discuss the proposed INO in this context.

# 2 The INO project

The INO is a proposed mega-science project that is jointly funded by the Department of Atomic Energy and the Department of Science and Technology, Government of India. The immediate goal is the creation of an underground (at least 1 km deep) cavern for scientific research purposes, with the main ICAL detector to study atmospheric neutrinos and cosmic muons. It will incorporate a centre for particle physics and detector technology at Madurai, South India, which currently houses the mini-ICAL prototype. The INO graduate programme trains students on both the theoretical and experimental aspects of neutrino/particle physics.

## 2.1 The ICAL detector

The proposed [7] 51 kton ICAL detector will consist of 151 layers of 56 mm thick soft iron that can be magnetised to about 15 kGauss. It will consist of three identical modules with the layers of iron plates separated by 40 cm air gaps in which the roughly 30,000 active Resistive Plate Chambers (RPCs) of dimensions $2 \times 2$ m$^2$ will be housed; see Fig. 1. The high DC voltage across the 3 mm glass plates separated by 2 mm of gas gap creates a discharge when a charged particle passes through them. These are digitised and stored as hits using the nearly 4 million channels of readout electronics, which are later analysed to reconstruct the kinematics of each event. While the minimum ionising muons leave clean long tracks in the detector, the hadrons shower. Hence the momentum, including direction, of the muons is well-reconstructed compared to that of the hadrons; in addition, the sign of the muon's charge can also be reconstructed to better than 98% for the few GeV events of interest.

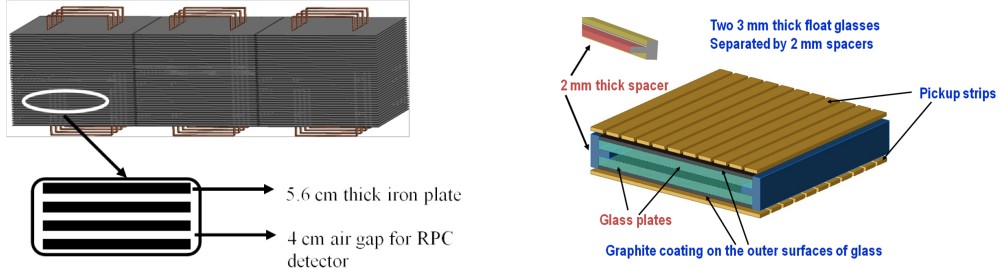

Figure 1: Schematic of ICAL detector and RPCs.

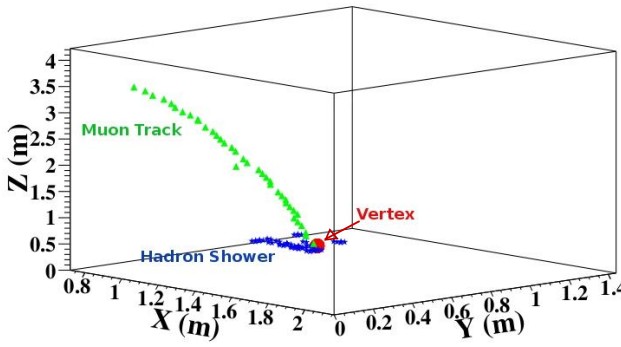

Figure 2: Typical event in ICAL simulations.

# 3 Atmospheric neutrinos and ICAL

Cosmic rays interact with the Earth's atmosphere to produce muons, which further decay to neutrinos, producing muon and electron neutrinos (and anti-neutrinos) in roughly 2:1 ratio. These neutrinos will mainly interact with the iron in the detector to produce charged particles in charged-current (CC) interactions:

$$
\begin{aligned}
\nu_\mu + N &\rightarrow \mu^- + X\,, \\
\overline{\nu}_\mu + N &\rightarrow \mu^+ + X\,,
\end{aligned}
\tag{2}
$$

where $X$ is any hadronic debris. The magnetic field bends the differently charged muons in different directions and hence differentiates between neutrino and anti-neutrino induced events. This separation allows for a sensitive study of matter effects which are different for neutrinos and anti-neutrinos and hence can pin down the neutrino mass ordering.

## 3.1 ICAL detector response

The physics reach of ICAL using detailed GEANT4 based simulations [8] is shown in this and the following sections. Unless specifically mentioned, all results are from the INO white paper [7]; we give the main highlights here.

The magnetic field in ICAL allows for a separation of $\mu^-$ and $\mu^+$ (neutrino or anti-neutrino induced) events as well as helps to determine the magnitude and direction of the muon's momentum. A typical atmospheric neutrino event is shown in Fig. 2. The lomg tracks of the mimimum ionising muons curve in the presence of the magnetic field and their momentum is reconstructed using a Kalman filter algorithm. The hadrons shower and appear as hits clustered close to the vertex. Typically 10,000 events with fixed momentum and direction were propagated in the simulated ICAL detector with GEANT, and their behaviour analysed.

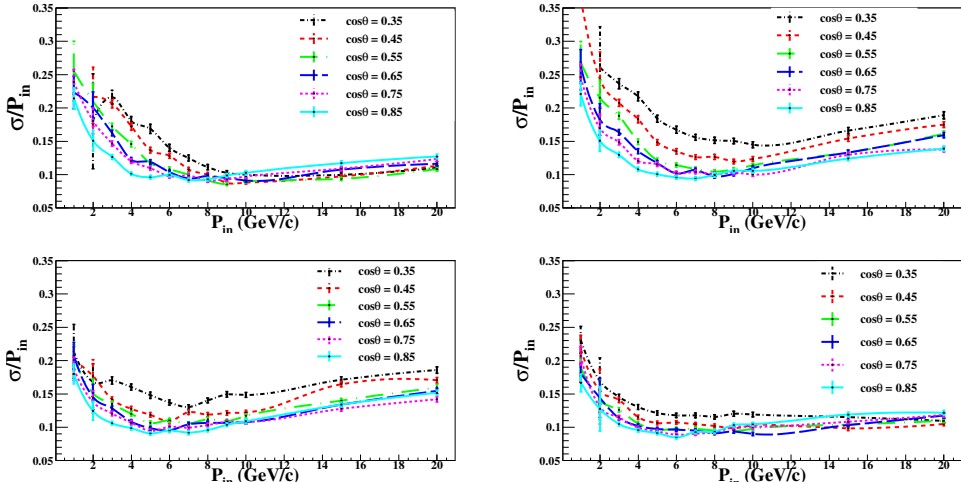

Figure 3: Muon resolution as a function of input momentum and $\cos\theta$, for different regions of azimuthal angles, $|\phi| \leq \pi/4$ (top left), $\pi/4 \leq |\phi| \leq \pi/2$ (top right), $\pi/2 \leq |\phi| \leq 3\pi/4$ (bottom left), $3\pi/4 \leq |\phi| \leq \pi$ (bottom right).

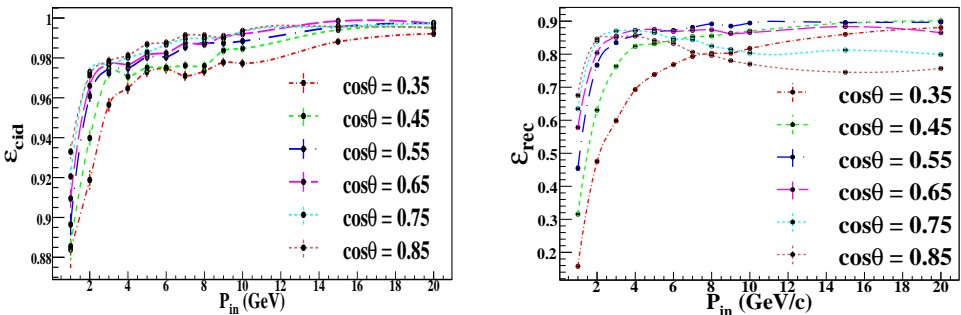

Figure 4: Reconstruction efficiency $\epsilon_{rec}$ (left) and charge-id efficiency $\epsilon_{cid}$ (right) as a function of the input momentum for different values of the zenith angle.

The muon momentum resolution (defined as $\sigma/P_{in}$, obtained by reconstructing fixed momentum muons in ICAL and fitting the resultant distribution to a gaussian distribution) is shown in Fig. 3 for different values of $(\cos\theta, \phi)$, with the upward direction corresponding to $\cos\theta = 1$. The azimuthal dependence occurs due to the presence of the magnetic field which breaks the azimuthal symmetry. (In addition, the angular resolution (for the zenith angle) is better than $1°$ for muons with more than $\approx 2$ GeV energy.)

Fig. 4 shows the reconstruction $\epsilon_{rec}$ and relative charge-id efficiency $\epsilon_{cid}$. The relative charge-id efficiency is the fraction of reconstructed events that were reconstructed with the correct charge sign. We see that $\epsilon_{cid} \gtrsim 98\%$ over a large range of momentum magnitude and direction.

Finally, the hadron energy resolution for fixed energy charged pion events is shown in Fig. 5. The final hadronic component in neutrino-iron interactions of interest at ICAL can contain multiple hadrons; however, pions dominate the sample; for details, please see Ref. [7].

## 3.2 Precision measurement of the 2–3 parameters

Honda atmospheric neutrino fluxes [9] were used to generate 1000 years of events using the ICAL geometry and the NUANCE neutrino generator [10] and then oscillated and scaled appropriately for the analysis. Typical events as a function of the muon zenith angle $\cos\theta$ are

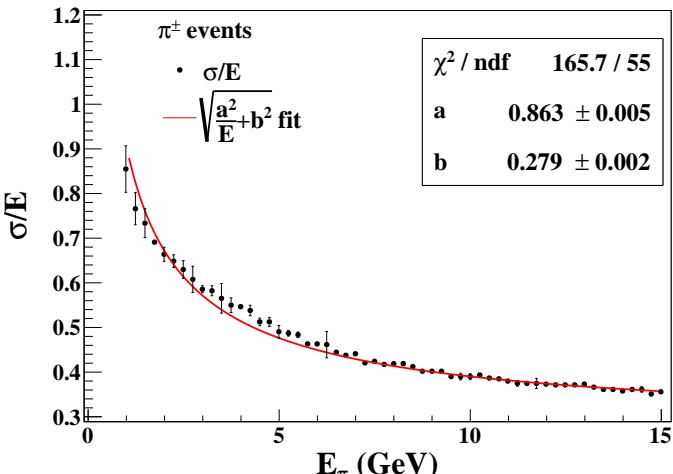

Figure 5: Hadron energy resolution as a function of pion energy. Here the reconstructed hadron energy was fitted to a Vavilov distribution and its resolution $\sigma/E$ computed. See Ref. [7] for details.

shown in Fig. 6 for events in the muon energy bin 2–3 GeV with an exposure time of 10 years. Finite detector resolutions smear these numbers somewhat. Notice the poor reconstruction in the horizontal direction.

Fig. 7 shows the precision that can be obtained in 10 years in the 2–3 parameters, $\sin^2\theta_{23}$ and $\delta_{32}$ for input values of $(0.5, 2.4 \times 10^{-3} \text{ eV}^2)$ in comparison with other experiments. Note that ICAL is yet to be built! The results are marginalised over the allowed $3\sigma$ range of $\sin^2\theta_{13}$ and include the reconstructed muon momentum and direction information, as well as hadron energy information. Results without hadron energy information are labelled 2D and those with hadron energy included are labelled 3D in the figure.

Fig. 7 also shows the sensitivity to the mass ordering: plotted is the minimum $\chi^2$ obtained when the data is fitted to the wrong (inverted) mass ordering as a function of the exposure time when the true ordering is normal. The result is marginalised over the magnitude of $\delta_{32}$ as well as $\theta_{23}$ and $\theta_{13}$. The inclusion of hadron information improves the result considerably. Similar results are obtained when the true hierarchy is inverted.

The neutrino mass ordering is the centrepiece of ICAL physics. Although many other experiments such as DUNE and JUNO will also measure this parameter, ICAL is complementary to these other experiments because of the separation of the neutrino and anti-neutrino events using the magnetic field. As a consequence, the sensitivity is independent of the CP phase $\delta_{CP}$, unlike in other experiments. Since this phase is currently unknown, the effects of this parameter must be disentangled from those due to the mass ordering in the other experiments. This can only be achieved for a fraction of the total range of $\delta_{CP}$. This feature of ICAL can also be used to obtain synergies with other experiments. For instance, the NO$\nu$A or ESS$\nu$SB sensitivity to the mass ordering depends significantly on $\delta_{CP}$; this sensitivity will be greatly improved on combining with ICAL data; see Fig. 8. The solid line shows the sensitivity of NO$\nu$A to the mass ordering: the "data" is generated with the normal ordering and then fitted assuming the inverted ordering, to obtain the minimum value of $\chi^2$. The results are plotted as a function of the true value of $\delta_{CP}$, and marginalised over all oscillation parameters, including $\delta_{CP}$. It is seen that NO$\nu$A has very little sensitivity to the mass ordering for values of $\delta_C < 180°$. Hence the reach of NO$\nu$A alone for the neutrino mass ordering is very sensitive to the actual value of $\delta_{CP}$. The sensitivity in this small-$\delta_{CP}$ region improves somewhat (dotted lines) on including reactor (especially T2K) data; and this improves dramatically (dashed lines) on including the ICAL information.

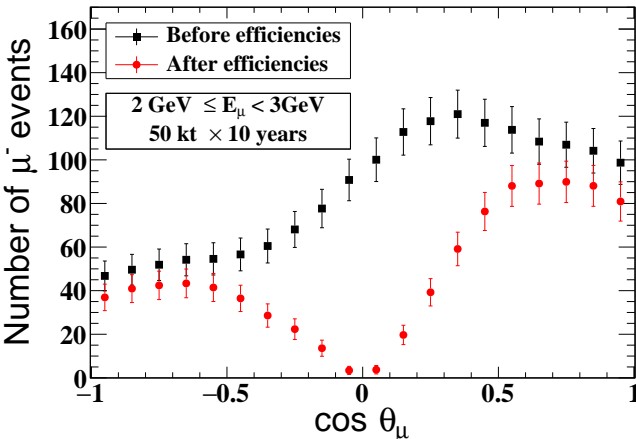

Figure 6: Zenith angle distributions of oscillated atmospheric muon $\mu^-$ events with energies from 2 to 3 GeV for an exposure time of 10 years. Statistical errors are shown. Note that there are about 2.5 times fewer $\mu^+$ events due to the smaller cross sections although their fluxes are approximately the same.

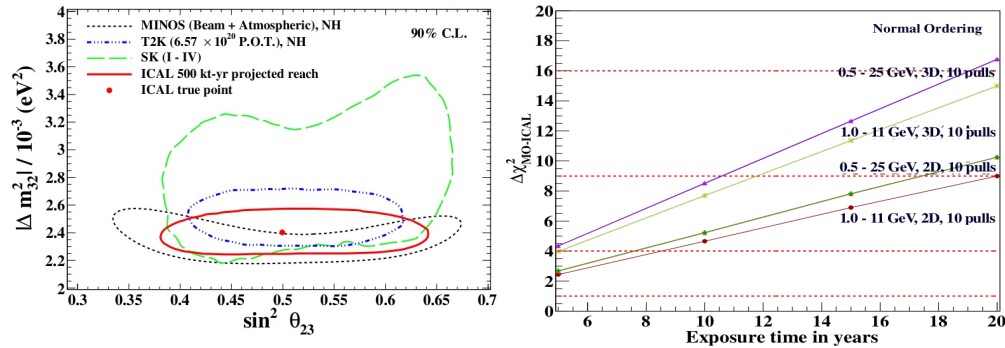

Figure 7: Left: Precision reach in 10 years of ICAL for $\sin^2\theta_{23}$ and $\delta_{32}$, including hadron energy information. Right: Sensitivity to the mass ordering for assumed input normal ordering as a function of exposure time; similar results are obtained for inverted ordering. The sensitivity increases with inclusion of larger muon energy range and inclusion of hadron energy information.

Fig. 8 also shows that inclusion of ICAL data will increase the sensitivity to $\delta_{CP}$ in regions of the $\delta_{CP}$ plane which earlier had poor sensitivity to this parameter.

## 3.3 Other measurements at ICAL

Electrons from CC interactions of electron neutrinos (and anti-neutrinos) will also produce showers in ICAL. It is not possible to separate the trackless events arising from neutral current interactions of all flavours of neutrinos in ICAL from these electron induced events. Low energy CC muon events where the muon track is not reconstructed are an additional contamination. It turns out that the trackless events also yield information [11] on the 2–3 neutrino mixing parameters, see Fig. 9. Here, the solid line indicates the sensitivity to $\sin^2\theta_{23}$ from just the trackless events at ICAL, when all other oscillation parameters are kept fixed. The inclusion of systematic uncertainties (dashed line) and further marginalising over the remaining oscillation parameters (dotted line) reduces the sensitivity, which remains significant.

Finally, charged tau are produced in CC interactions of tau neutrinos with the detector. Since atmospheric tau neutrinos are suppressed by a factor of about $10^5$ compared to atmo-

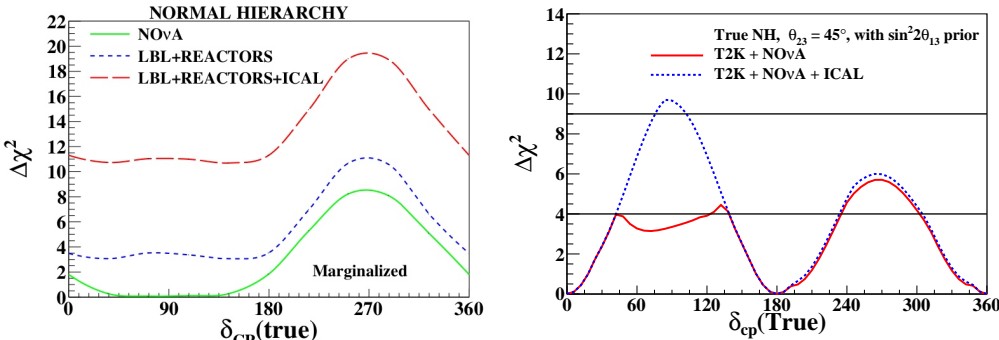

Figure 8: Left: Synergies with other experiments that improve the determination of the mass ordering over the entire $\delta_{CP}$ range. Here the full proposed runs of the long baseline and reactor experiments are taken, with 10 years' running of ICAL. Right: Synergies that improve the determination of $\delta_{CP}$.

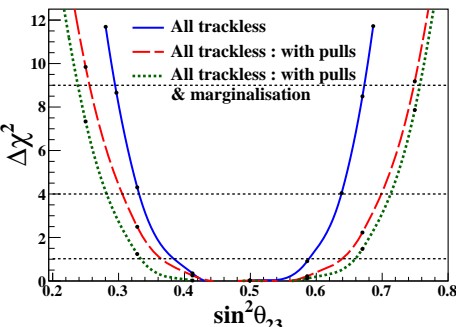

Figure 9: Sensitivity to $\sin^2 \theta_{23}$ from a simulations analysis of "trackless" events. See text for details.

spheric muon neutrinos, significant yield of tau neutrino events are a direct signature of neutrino oscillations, dominantly of the atmospheric muon neutrinos. A simulations study of the tau neutrino-induced CC events at ICAL, with the resulting taus decaying hadronically [12], are also sensitive to neutrino oscillations. Since the threshold for CC tau production is 3.5 GeV, such events are visible as an excess over the neutral current (NC) background although they cannot be classified event-by-event. An analysis of the combined NC and CC$\tau$ events shows sensitivity to both the 2–3 oscillation parameters $\theta_{23}$ and (the magnitude of) $\Delta m_{31}^2$. Since CC muon events arise from the *same* atmospheric neutrino fluxes, several systematic uncertainties are the same, and in fact, simulation studies show that a combined study of muon and tau CC events improves the sensitivity to the 2–3 oscillation parameters, in particular, the unknown octant of the mixing angle $\theta_{23}$, as seen in Fig. 10.

There are many other studies of the sensitivity of ICAL to cosmic ray muons, to BSM physics such as neutrino decays and sterile neutrinos, non-standard interactions and Lorentz invariance violations, and the ability to search for dark matter (WIMPS) and exotics such as monopoles.

## 4 Prototypes and the mini-ICAL detector

Cosmic muons have been observed at several RPC stacks that have been built for calibration and testing. One such stack at Madurai in South India has measured [13] the cosmic muon

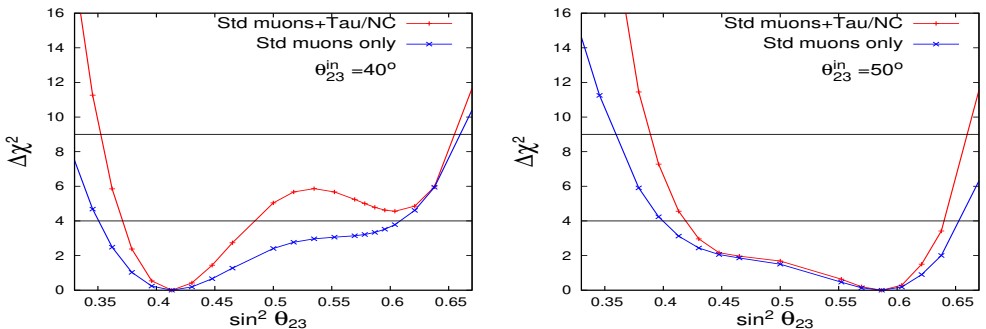

Figure 10: Sensitivity to the octant of $\sin^2 \theta_{23}$ from a combined simulations analysis of CC muon and tau events, for an input value of $\sin^2 \theta_{23} = 40°$ (left) and $50°$ (right).

flux as a function of both the zenith angle and the azimuthal angle and has also measured the east-west asymmetry of cosmic ray muons; see Fig. 11.

The mini-ICAL detector; see Fig. 12, is a $4 \times 4$ m$^2$ 85 ton scaled model with 11 iron layers that has been constructed two years ago. After testing and calibration, the cosmic muon

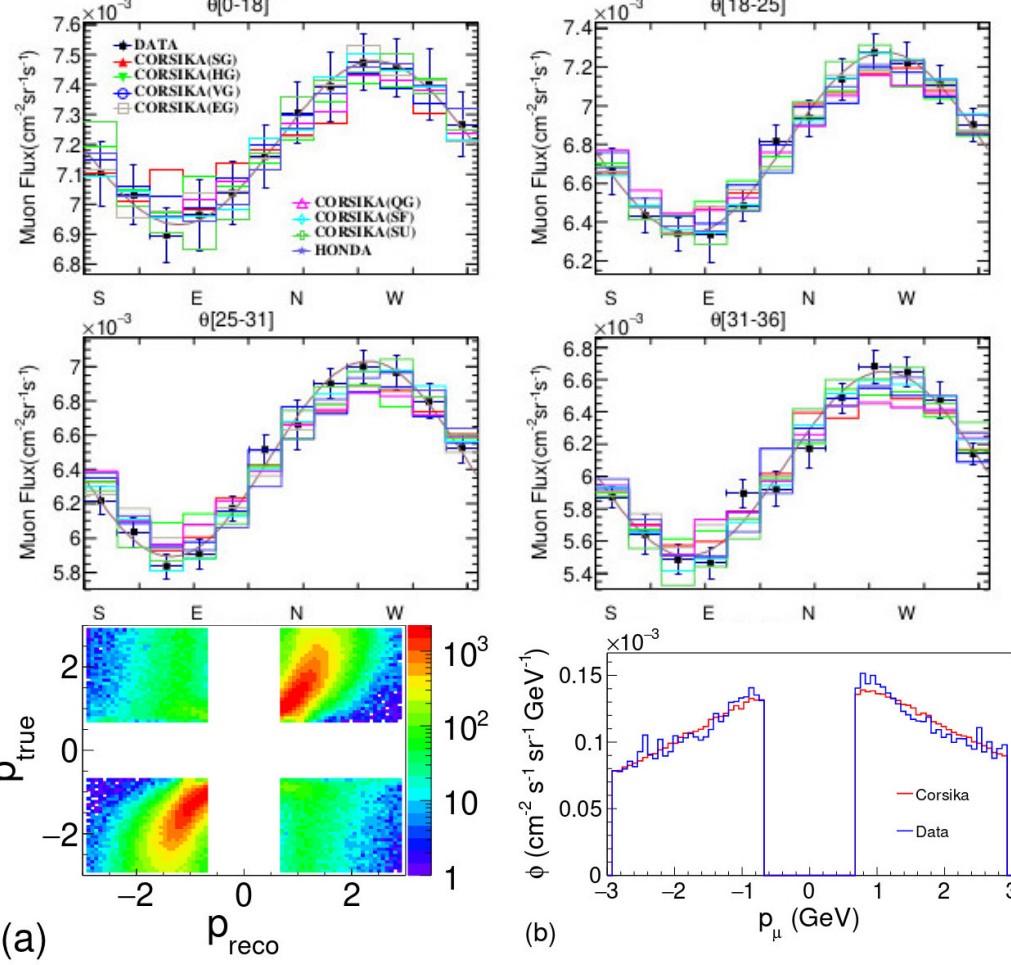

Figure 11: Left: Measured east-west asymmetry of cosmic ray muon flux at the Madurai RPC stack. Right: Cosmic muon spectrum from the mini-ICAL detector at Madurai.

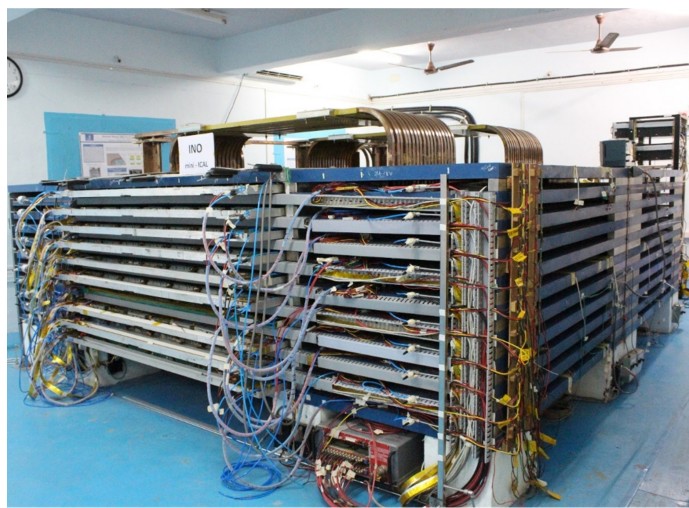

Figure 12: The prototype mini-ICAL detector at Madurai, South India.

spectrum has been measured with this detector and is consistent with theoretical expectations; in addition, the $\mu^-$ and $\mu^+$ events have been separately observed [14] and are shown as the spectra with positive and negative momenta respectively in Fig. 11.

## 5 Conclusion

Simulations studies show that ICAL has excellent physics potential, especially to precisely measure the 2–3 neutrino mixing parameters as well as the neutrino mass ordering. Detailed studies of charged current muon events arising from interactions of atmospheric muon neutrinos have been performed; studies of CC electron neutrino and tau neutrino events are ongoing. Separately, simulations studies have also examined the sensitivity of ICAL to beyond-the-standard-model (BSM) physics, exotic particles such as WIMPS and monopoles, etc. ICAL will be complementary in its reach to other detectors around the world such as DUNE and JUNO. Finally, ICAL is a completely indigenous detector with excellent R&D on RPC design and construction including closed-loop gas system and automation by industry. In addition, the detector electronics (analog and digital front end), including chip design and development is also complete. A 700 ton large scale $8 \times 8 \times 2$ m$^3$ engineering module is also to be constructed shortly.

In parallel, research is going on into neutrinoless double beta decay experiments using tin bolometry, and dark matter experiments, which are also proposed to be housed at INO. While it is expected that INO, when built, will galvanise students as well as local industry due to cutting-edge technology transfer, there have been delays in obtaining some technical clearances for the underground lab construction. The INO collaboration is looking forward to going ahead soon with this ambitious project.

## Acknowledgements

I thank the conference organisers and the members of the INO collaboration for this opportunity to present the physics reach and status report of INO.

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
