# Peer review of "India Based Neutrino Observatory, Physics Reach and Status Report"

_SciPost Physics Proceedings, doi:SciPost Phys. Proc. 13, 024 (2023)_

## Round 1 · Referee Report · Anonymous (Referee 1) · 2022-10-15

Report

Comments - Beginning on page 4, clarify if the MO sensitivity plot (fig. 2) does include hadron information. - Fig. 3 could use a more detailed caption. Is it showing sensitivity to delta-CP for 10 years? - Fig. 4 it's not clear what the different lines "with pulls", "with pulls and marginalization" indicate. - Sec. 3 would benefit from showing an event display and/or event distributions from the simulation studies. - What is the energy sensitivity expected? tau detection has a threshold of ~2GeV. Is ICAL expecting to be able to classify taus event by event? - In the conclusion, "pending litigation" sounds ominous... perhaps can soften a bit.

Minor corrections - "aout 2 eV" --> "about" - "INO grauate" --> "graduate" - "interact with mainly " --> "mainly interact with"

---

## Round 2 · Referee Report · Anonymous · 2022-10-26

Report

All my comments have been addressed and the proceeding is suitable for publication.

---

## Round 2 · Author Response

We thank the referee for very pertinent and useful comments. We have made all the changes as suggested as well as have added a subsection with more details on detector response.

---

## Round 2 · List of Changes

Response to referee comments on 2022-10-15

We thank the referee for very pertinent comments which have greatly improved the quality of the write-up. Detailed responses to each comment appear below.

1. - Beginning on page 4, clarify if the MO sensitivity plot (fig. 2) does include hadron information.

Response: Yes it does. This has been clarified in the text and the figure itself (now labelled Fig 7) replaced for clarity.

2. - Fig. 3 could use a more detailed caption. Is it showing sensitivity to delta-CP for 10 years?

Response: This was a mistake on my part. I have corrected both the caption and the figures as follows: in the figure, which is now labelled Fig 8, the orginial left figure has been removed since it was technical and not needed. The right hand figure (now left hand) is now explained in the text and actually refers to the improvement in mass ordering and not $\delta_{CP}$ as stated earlier. The new right hand figure now refers to the improvement in $\delta_{CP}$. Appropriate text has been included.

3. - Fig. 4 it's not clear what the different lines "with pulls", "with pulls and marginalization" indicate.

Response: This has been clarified. The figure is now labelled Fig 9 and details appear in the text.

4. - Sec. 3 would benefit from showing an event display and/or event distributions from the simulation studies.

Response: Fig 2 showing a typical event has been included.

5. - What is the energy sensitivity expected? tau detection has a threshold of ~2GeV. Is ICAL expecting to be able to classify taus event by event?

Response: An entire section 3.1 on ICAL detector response has been included, along with Figs 3, 4, 5. Sample events that will be obtained with 10 years exposure at ICAL are shown in Fig. 6. References for Honda atmospheric neutrino fluxes, NUANCE neutrino generator and GEANT4 have been added. The expected muon and hadron energy sensitivities are clearly shown, along with reconstruction and charge-id efficiencies. Tau production through CC process has a threshold of $E_\nu > 3.5$ GeV. When taus decay hadronically, they appear as an excess over neutral current events and are statistically significant as the NC events fall off sharply with energy. Added explanations have been included in the appropriate part of the text in Page 7. The taus appear as a statistical excess and cannot be separated event by event.

6. - In the conclusion, "pending litigation" sounds ominous... perhaps can soften a bit.

Response: Thanks for this feed-back; the sentence has been altered to
``While it is expected that INO, when built, will galvanise students as well as local industry due to cutting-edge technology transfer, there have been delays in obtaining some technical clearances for the underground lab construction."

7. - Minor corrections
- "aout 2 eV" --> "about"
- "INO grauate" --> "graduate"
- "interact with mainly " --> "mainly interact with"

Response: Thanks for such a careful reading: all typos have been corrected.

We hope that the write-up is now suitable for publication.

---

## Editorial Decision

published